# Gene-Delivery Ability of New Hydrogenated and Partially Fluorinated *Gemini bispyridinium* Surfactants with Six Methylene Spacers

**DOI:** 10.3390/ijms23063062

**Published:** 2022-03-11

**Authors:** Michele Massa, Mirko Rivara, Gaetano Donofrio, Luigi Cristofolini, Erica Peracchia, Carlotta Compari, Franco Bacciottini, Davide Orsi, Valentina Franceschi, Emilia Fisicaro

**Affiliations:** 1Department of Maternal Infantile and Urological Sciences, Sapienza University of Rome, 00165 Rome, Italy; michele.massa@uniroma1.it; 2Department Food and Drug, University of Parma, Parco Area delle Scienze 27/A, 43124 Parma, Italy; mirko.rivara@unipr.it (M.R.); erica.peracchia@studenti.unipr.it (E.P.); carlotta.compari@unipr.it (C.C.); franco.bacciottini@unipr.it (F.B.); 3Department of Veterinary Sciences, University of Parma, Via del Taglio 10, 43126 Parma, Italy; gaetano.donofrio@unipr.it (G.D.); valentina.franceschi@unipr.it (V.F.); 4Department of Mathematical, Physical and Computer Science, University of Parma, Parco Area delle Scienze 7/a, 43124 Parma, Italy; luigi.cristofolini@unipr.it (L.C.); davide.orsi@unipr.it (D.O.)

**Keywords:** heterocyclic *gemini* cationic surfactants, nonviral vectors, gene delivery, gene therapy, partially fluorinated *gemini* surfactants, atomic force microscopy on DNA, DNA-surfactant interaction

## Abstract

The pandemic emergency determined by the spreading worldwide of the SARS-CoV-2 virus has focused the scientific and economic efforts of the pharmaceutical industry and governments on the possibility to fight the virus by genetic immunization. The genetic material must be delivered inside the cells by means of vectors. Due to the risk of adverse or immunogenic reaction or replication connected with the more efficient viral vectors, non-viral vectors are in many cases considered as a preferred strategy for gene delivery into eukaryotic cells. This paper is devoted to the evaluation of the gene delivery ability of new synthesized *gemini* bis-pyridinium surfactants with six methylene spacers, both hydrogenated and fluorinated, in comparison with compounds with spacers of different lengths, previously studied. Results from MTT proliferation assay, electrophoresis mobility shift assay (EMSA), transient transfection assay tests and atomic force microscopy (AFM) imaging confirm that pyridinium *gemini* surfactants could be a valuable tool for gene delivery purposes, but their performance is highly dependent on the spacer length and strictly related to their structure in solution. All the fluorinated compounds are unable to transfect RD-4 cells, if used alone, but they are all able to deliver a plasmid carrying an enhanced green fluorescent protein (EGFP) expression cassette, when co-formulated with 1,2-dioleyl-sn-glycero-3-phosphoethanolamine (DOPE) in a 1:2 ratio. The fluorinated compounds with spacers formed by six (FGP6) and eight carbon atoms (FGP8) give rise to a very interesting gene delivery activity, greater to that of the commercial reagent, when formulated with DOPE. The hydrogenated compound GP16_6 is unable to sufficiently compact the DNA, as shown by AFM images.

## 1. Introduction

Gene therapy is based on the transfer of genetic material into cells to treat an inherited or acquired disease, or at least to improve the clinical status of a patient, or to prevent a disease. With the rapid conclusion of the Human Genome Sequencing Project, the idea has matured to treat diseases caused by a known genetic defect by delivering to the diseased cells or body organs a correct copy of the defective gene. This is the rationale of gene therapy, now a clinical reality, after the approval by FDA of the in vivo treatment of ADA-SCID and Leber congenital amaurosis, or ex vivo CAR-T therapy for treating B-lymphoid malignancies [1,2]. The focus of cancer treatment has in fact shifted towards highly personalized targeted therapies as immunotherapy, therapies that engage and reinforce the power of a patient’s immune system against tumors. Gene-based therapeutics use many different approaches as for instance gene silencing or activation of suicidal genes, ribozymes, aptamers and small interfering RNAs [1,2,3,4]. Notwithstanding most of the approved gene therapy clinical trials has the aim to fight cancer [5], due to the pandemic emergency determined by the spreading worldwide of the SARS-CoV-2 virus, the possibility to prevent infectious diseases by genetic immunization has focused the attention of the pharmaceutical industry and of the governments [6,7,8,9,10,11]. The only way to reach an effective herd immunity to mitigate the adverse effect of the virus on public health, economy and social relationships is the availability of a safe and effective vaccine. Since the genetic sequence of the virus was completed in early January 2020, the development of genetically based vaccines has moved forwards at unpredictable speed [11]. Because it has been shown that antibodies targeting the spike protein are able to neutralize the virus [12,13], gene-based vaccines are currently the most used around the world. DNA vaccines consist of plasmid DNA encoding the spike gene under a mammalian promoter; RNA vaccines are based on RNA encoding the spike protein. The expression of the protein can take place after the delivery of the genetic material inside mammalian cells. This path is impossible for unprotected genetic material, degraded by the endo- and exonucleases present in physiological fluids. Moreover, the large size with high negative charge and the strong hydrophilicity are further barriers limiting the delivery [3]. Whereas in RNA vaccines the genetic material is packaged in lipid nanoparticles, the DNA ones use viral vectors because they need more efficient delivery devices not only to overcome the cell membrane, but also to reach the nucleus for expression.

Regardless of the kind of genetic material to be introduced inside the cells, the major issue that gene therapy strategy is facing concerns the gene delivery systems (known also as vectors). Most of the vectors used in gene therapy fall into two classes, namely viral or non-viral. Due to the risk of adverse or immunogenic reaction or replication connected with the more efficient viral vectors, non-viral vectors are in many cases considered as a preferred strategy for gene delivery into eukaryotic cells. It is therefore evident that the development of very efficient and safe vectors is a landmark for gene-based medicine. Sharma and Cow [3] have very recently reviewed the progress in nonviral gene delivery. The reader should refer to their work for a more complete treatment of this subject.

One of the advantages coming from synthetic vectors is the possibility to have very reproducible and controlled substances, a goal more difficult to achieve with viral vectors. Moreover, the chemical synthesis offers the possibility to understand the effect of the various moieties constituting the molecule for optimization purposes and, eventually, to add to the vector a chemical group that can be recognized by the target cell. This is the area in which our research is moving. 

For many years, in fact, we have been studying from synthetic, thermodynamic, and biomedical point of view, new cationic lipids with the aim to determine structure–activity relationships useful for the optimization of their gene delivery ability [14,15,16,17,18,19,20,21]. Particularly, we devoted our attention to the *gemini* cationic surfactants—i.e., surfactants consisting of two or more identical hydrophobic chains and two or more polar positive head groups covalently bound together by a spacer. *Gemini* surfactants have been attracting increased attention from the end of the last century, due to their enhanced surface properties (increased surface activity, low critical micelle concentration (cmc), useful viscoelastic properties) with respect to the monomeric counterparts [22,23,24,25,26,27,28]. Taking advantage of these properties, they have been proposed as components of new drug delivery systems and as non-viral vectors for gene delivery [29]. Their multiple cationic character allows the binding and compacting of DNA into soft nanoparticles of tunable size, able to protect the DNA from enzymatic degradation and to prevent rapid leakage into blood capillary but small enough to escape macrophages of the reticuloendothelial system [30,31,32,33,34,35,36,37,38,39]. Biomedical and biopharmaceutical applications of cationic amphiphiles as gene delivery vectors, together with their chemico-physical and aggregation properties, have been collected in a very interesting book [28]. *Gemini* surfactants with heterocyclic polar heads and their use as nonviral vectors have been extensively reviewed [29,31]. The use of *gemini* surfactants as candidates for gene delivery and the effect on the efficiency of transfection of the chemical structure of the surfactant (variations in the alkyl tail length and spacer/head group) and of the resulting physicochemical properties of the lipoplexes was also emphasized [34]. In the following, we examine the effect of the spacer length having in mind the optimization of the delivery ability of both hydrogenated and partially fluorinated dipyridinium *gemini* surfactants with chlorides as counterions (GP16_*n* and FGP*n*, respectively, where *n* is the number of methylene groups in the spacer). The interest in fluorinated surfactants is due both to their chemical and biological inertness and to their hydrophobic and at the same time lipophobic character. When therapeutic genes are delivered through biological fluids containing endogenous hydrogenated interfering surfactants, as pulmonary surfactants or bile salts, fluorination could prevent the lipoplexes being destroyed before entering inside the diseased cells [40,41,42]. Fluorination of cationic lipids has been proposed, for instance, in the treatment of cystic fibrosis and cystic fibrosis-associated diseases [40].

## 2. Results

### 2.1. Synthesis

We synthesized two novel *gemini* pyridinium surfactants having a 6 methylene group as a spacer between the cationic heads (see Figure 1). The two new compounds here reported have attached to the cationic heads two different chains: one is an alkyl C16 hexadecane and the other a fluorinated 3,3,4,4,5,5,6,6,7,7,8,8-tridecafluorooctane. We prepared the 1,6-bis(2-pyridyl)hexane and the two new surfactants following and adapting, when necessary, the procedures previously reported [15,19,20]. The synthetic scheme is reported in supporting information (SI). All the newly synthesized compounds are >95% pure by elemental analysis and by NMR spectra (all characterization data are available in Appendix A).

### 2.2. EMSA and MTT Assays

EMSA and MTT assays were carried out to find out the optimum conditions for transient transfection experiments. We started by testing the cytotoxicity of the molecules by an MTT proliferation assay to find out the concentrations useful for gene delivery purposes. Results are shown in Figure 2 for the GP16_6 and in Figure 3 for FGP6 in comparison with FGP8, the last compound showing the greatest gene delivery ability between the partially fluorinated compounds previously studied. 

For GP16_6, the cytotoxicity is a little greater on RD4 cells than on A549 cells. Above 10 μM, the viability of the cells decreases significantly. Fluorinated compounds are more toxic, and, at a concentration greater than 5 μM, the viability of all RD4 and A549 decreases abruptly, whereas LA4 survives till 10 μM. We have shown that the cytotoxicity of the partially fluorinated compounds seems to increase with the length of the spacer and, for the highest concentration tested, also in the presence of DOPE [21].

From EMSA and MTT results, we have chosen the 5 μM concentration for the transfection experiments as the best compromise between cell viability and transfection efficiency, taking into consideration that cytotoxicity has a strong impact on cell transfection. Another interesting piece of information about the ability of the compounds under investigation to interact with DNA is given by EMSA experiments (Figure 4). Whereas fluorinated compounds are able to interact and compact DNA starting from a *N*/*P* = 1/1 ratio (100 μM), the hydrogenated one shifts the DNA only at a concentration of 200 μM.

### 2.3. AFM and DLS

To better understand the interaction between the compounds under study and the DNA from a morphological point of view, the AFM images, obtained in the same conditions as transient transfection experiments, were obtained. AFM experiments were carried out in air in tapping mode using circular DNA, as described in the experimental section. 

First of all, the plasmid DNA alone deposited onto freshly cleaved mica was imaged (Figure 5a). Single plasmids and concatamers are seen in their plectonemic form, well extended all over the mica surface, with several supercoils that cause the double helix to cross itself a few times. The images in the presence of the cationic surfactants were taken at a *N*/*P* ratio = 1, the same condition used in transfection experiments. 

The compound GP16_6, in the conditions adopted, is unable to compact DNA in nanoparticles, as suggested also by EMSA results (Figure 5b). On the contrary, partially fluorinated compound can condense all the DNA in small near spherical nanoparticles of quite uniform size (Figure 5c,d), probably of the right dimension for delivering DNA inside cells. Moreover, some quite big particles (not shown) are also found on the mica disk. For the smaller particles shown in Figure 4c, a medium diameter of around 50 nm can be evaluated.

These results are confirmed by DLS experiments: in panel Figure 5e, we report the volume distribution obtained from a freshly prepared sample of plasmid and FGP6. This distribution (PDI = 0.60, where PDI is the polydispersity index) features a first peak with a maximum at 56 nm and standard deviation of 8 nm, together with a broader maximum at a larger size, ranging up to a few hundreds of nm. While the first peak is presumably related to lipoplexes involving individual DNA molecules, the broader maximum can be attributed to aggregates, as it develops and grows in size with time (data not shown). The formation and growth of larger clusters is indicative of an incipient aggregation process. We also measured the zeta potential on a fresh sample, finding ζ = −48 ± 1 mV. Similar results can be found in literature [43,44,45].

### 2.4. Transient Transfection Experiments

The ability of the compounds under investigation to deliver DNA inside the cells was studied by a transient transfection assay. Results on RD4 (left) and A549 cells (right) by GP16_6, FB6 and FB8 5 μM are shown in Figure 6, where, on the left, phase contrast and, on the right, fluorescence microscope observation of the transfected cells (as shown by green cells expressing EGFP) are reported. The experiments, done with the surfactant alone and with surfactant:DOPE = 1:2, point out that the presence of DOPE is needed for transfection. Whereas GP16_6 is unable to give transfection both in the absence and in the presence of DOPE, the fluorinated compounds show, in the presence of DOPE, a very high transfection ability superior to that of a commercial reagent, used as reference.

## 3. Materials and Methods

### 3.1. Compounds

The compounds under study (GP*m_n* and FGP*n*, where *n* = 3, 4, 6, 8, 12 indicates the number of carbon atoms in the spacer and *m* the length of the hydrophobic chains of hydrogenated surfactants) were prepared by us. 

The general IUPAC names of the compounds studied are 1,1′-bis-hexadecyl-2,2′-*n*-methylenebispyridinium dichloride (GP16_*n*) and 1,1′-bis(3,3,4,4,5,5,6,6,7,7,8,8,8-tridecafluorooctyl)-2,2′-*n*-methylenebispyridinium dichloride (FGP*n*). Their structures are shown in Figure 1. 

#### 3.1.1. Chemical Synthesis 

Two novel *gemini* pyridinium surfactants having the 6 methylene group as a spacer between the cationic heads were synthesized following and adapting, if necessary, the procedures previously reported [15,19,20]. The NMR spectra were recorded using a Bruker 300 Avance (Billerica, MA, USA) spectrometer and a Bruker 400 Avance (Billerica, MA, USA) spectrometer.

^1^H NMR Spectra (300 MHz) chemical shifts (δ scale) are reported in parts per million (ppm) and are also reported in order multiplicity, and number of protons. Signals were characterized as s (singlet), d (doublet), t (triplet), m (multiplet), and br s (broad signal).

^13^C NMR (100 MHz); chemical shifts (δ scale) are reported in parts per million (ppm). 

^19^F-NMR (376.5 MHz); chemical shifts (δ scale) are reported in parts per million (ppm).

IR spectra were recorded using a Agilent Technologies Cary 630 FTIR Spectrometer (Santa Clara, CA, USA) in the region 700–4000 without KBr support. 

Mass spectra were recorded using an Applied Biosystem/MDS SCIEX API-150 EX (Waltham, MA, USA) instrument.

The new compounds were analysed on a ThermoQuest (Rodano, Italy) FlashEA 1112 Elemental Analyzer, for C, H, *N*. The percentages recorded were within 0.4% of the theoretical values. 

##### Synthesis of the 1,6-Bis(2-pyridyl)hexane

In a dried three necked round-bottom flask, equipped with gas (nitrogen), anhydrous diethyl ether (37.5 mL) and butyllithium (12.5 mL 0.02 mol (1.6 M solution in hexane)) were introduced. The temperature was lowered to −20 °C with a mixture of ice and CaCl_2_ hexahydrate and then the 2-methylpyridine (0.02 mol) was added dropwise, under stirring. Dibromobutane (0.01 mol) was added dropwise, over a period of 20 min, keeping the temperature between −25 and −20 °C. The reaction was kept at −20 °C for 50 min and then allowed to return to room temperature and stirred for an additional 2 h. The reaction was quenched with water. The ethereal solution was extracted with HCl (6 N). The aqueous layer was separated and treated with a sodium hydroxide aqueous solution until the pH was basic. The aqueous phase was extracted three times with diethyl ether and the organic phase was dried over sodium sulphate and evaporated under reduced pressure. The crude product was purified with Flash Chromatography on a silica gel column using a gradient starting with methylene chloride and finishing with ethyl acetate obtaining a pale-yellow oil. Yield 1.62 g (68%); ^1^H-NMR (300 MHz CDCl_3_): δ 1.20 (m, 4 H); δ 1.57 (m, 4 H); δ 2.57 (t, 4 H); δ 6.88 (d 4 H); δ 7.35 (td, 2 H); δ 8.30 (d, 2 H) (see Supporting Information, SI).

##### Synthesis of 1,1′-Bis(3,3,4,4,5,5,6,6,7,7,8,8,8-tridecafluorooctyl)-2,2′-hexamethylenebis (Pyridinium) Chloride

Into a dried three necked round-bottomed flask, 1,1,1,2,2,3,3,4,4,5,5,6,6-tridecafluoro-8-(methylsulfonyl)octane (0.0055 mol) was introduced after being diluted with 25 mL of dichloromethane; then, the temperature was increased to 60 °C and the 1,6-bis(2-pyridyl)hexane) (0.0025 mol), diluted in dichloromethane (10 mL), was added dropwise. The reaction was stirred at the same temperature overnight. The reaction then cooled with an ice bath, and the product was suspended a few times in ethyl acetate. The crude product was then submitted to ion exchange procedure using (Amberlite^®^ IRA400 from Aldrich, St. Louis, MO, USA) following the already published procedure [15] to obtain the desired product in its chloride form as a yellow hard oil. Yield 0.96 g (40%); ^1^H-NMR (300 MHz CDCl_3_): δ 1.71 (m, 4 H); δ 1.96 (m, 4 H); δ 3.28 (t, 8 H); δ 5.10 (t, 4 H); δ 8.06 (td, 2 H); δ 8.19 (d, 2 H); δ 8.61 (td, 2 H); δ 9.12 (d, 2 H). ^13^C NMR (100 MHz, MeOD): δ 154.5, 146.8, 145.2, 130.3, 127.1, 58.3, 32.1, 29.5, 28.7, 26.2. ^19^F NMR (376.5 MHz, [D_6_]DMSO): δ = −80.67 (3F), −113.61 (2F), −122.01 (2F), −123.55 (4F), –126.86 (2F) ppm. FT-IR: *n* = 3377, 2997, 2858, 1632, 1513, 1453 1233, 1189, 1140, 1021, 782, 697 cm^−1^ (see SI). MS-ESI: *m*/*z* = 969 [M-Cl]. Anal. Calcd. For C_32_H_28_Cl_2_F_26_N_2_ (1004.12): C 38.23, H 2.81, *N* 2,79. Found C 38.30, H 2.78, *N* 2.73.

##### Synthesis of 1,1′-Bis-hexadecyl-2,2′-hexamethylenebispyridinium Chloride (GP16_6)

Into a three necked round-bottomed flask, the 1,6-bis(2-pyridyl)hexane (1.62 g, 0.007 mol) was dissolved in DMF (10 mL). The temperature was increased to 150 °C, and the hexadecyl chloride (17.6 g, 20.25 mL, 0.07 mol) was slowly added to the solution. The reaction stirred overnight and the DMF was evaporated under vacuum. The residue was suspended in diethyl ether and crystallized two times from acetonitrile/toluene giving yellow-gray crystals. Yield 3.23 g (63%); ^1^H-NMR (300 MHz MeOD): δ 0.92 (t, 6 H); δ 1.35 (m, 44 H); δ 1.76 (m, 8 H); δ 2.14 (m, 8 H); δ 3.15 (m, 4 H); δ 3.37 (d, 4 H); δ 5.09 (t, 4 H); δ 802 (t, 2 H); δ 8.20 (d, 2 H); δ 8.59 (t, 2 H); δ 9.06 (d, 2 H). ^13^C NMR (100 MHz, MeOD): δ 153.9, 146.8, 128.3, 124.9, 56.9, 25.1, 24.7, 23.8, 21.2, 21.1, 20.2, 19.8, 18.1, 15.0, 13.9. FT-IR: ν = 3373, 3041, 2922, 2851, 1625, 1513, 1464, 1170, 793, 723 cm^−1^ (see SI). MS-ESI: *m*/*z* = 725 [M-Cl]. Anal. Calcd. For C_48_H_86_Cl_2_N_2_ (760.62): C 75.65, H 11.37, *N* 3.68. Found C 75.59, H 11.32, *N* 3.72.

### 3.2. DNA Preparation and Storage

Plasmid DNA was purified through caesium chloride gradient centrifugation as reported in [14,18,21]. A stock solution of the plasmid 0.7 μM in milliQ water (Millipore Corp., Burlington, MA, USA) was stored at −20 °C. Linearized plasmid DNA (pEGFP-C1, Clontech, Takara Bio USA Inc., Mountain View, CA, USA) was obtained by cutting with EcoRI restriction enzyme (Roche, Monza, Italy), column purified (Genomed, Leesburg, FL, USA) and precipitated in alcohol. The pellet of the linearized plasmid DNA was dissolved in distilled water at a final concentration of 1 μg/μL, after washing with 70% of ethanol and air drying. 

### 3.3. Cell Culture

In our experiments, we used the human rhabdomyosarcoma cell line RD-4 (ATCC^®^ CCL-136™), obtained from David Derse, National Cancer Institute, Frederick, MD, USA, the human pulmonary adenocarcinoma cell line A549 (ATCC^®^ CCL-185™) and the murine pulmonary adenoma (Mus musculus Lung adenoma) cell line LA-4 (ATCC^®^ CCL-196™). All the cell lines were maintained as a monolayer using growth medium containing 90% complete EMEM (cEMEM, Eagle’s minimal essential medium, Gibco; Thermo Fisher Scientific Carisbad, CA, USA) completed with 2 mM L-glutamine, 100 IU/mL penicillin, 10 µg/mL streptomycin, 2.5 μg/mL of amphotericin B and 1 mM of sodium pyruvate, all purchased from Gibco) and 10% FBS (Fetal Bovine Serum, Gibco). Cells were subcultured to a fresh culture vessel when growth reached 70–90% confluence (i.e., every 3–5 days) and incubated at 37 °C in a humidified atmosphere of 95% air/5% CO_2_ [14,18,21]. 

### 3.4. Electrophoresis Mobility Shift Assay (EMSA)

In addition, 14 μL of several different final concentrations of surfactants, ranging from 6.5 to 200 μM, were obtained by adding to 25 μL of 20 mM Tris/HCl buffer at pH 8, 3 μL of GP16_6, FGP6 and FGP8 (all stored at 2 mM) after twofold serial dilutions. Furthermore, 1 μg of pEGFP-C1 plasmid DNA (Clontech) was then added and binding reaction was allowed to take place at room temperature for 30 min. The reactions were loaded on a TA 1× (Tris base (2-Amino-2-hydroxymethyl-propane-1,3-diol 40 mM and acetic acid 20 mM) 1% agarose gel. The gel was run for 1 h in TA buffer at 10 V/cm, and EDTA was omitted from the buffers because it competes with DNA in the reaction [14,18,21].

### 3.5. MTT Proliferation Assay

RD-4, A549 and LA-4 cells were plated on 96-well plates (4000 cells per well). GP16_6, FGP6 and FGP8 were added at several concentrations (2.5, 5, 10, 20 and 40 μM). The same experiments were carried out also by adding 1,2-dioleyl-sn-glycero-3-phosphoethanolamine (DOPE; SIGMA-Aldrich, St. Louis, MO, USA) at a surfactant:DOPE ratio of 1:2. At 24 h post treatment, the relative number of metabolically active cells was assessed by reduction of 3-[4,5-dimethylthiazol-2-yl]-2,5-diphenyl tetrazolium bromide (MTT). 

The assay was performed as described in [14,18,21]. Briefly, after 24 h of treatments, 10 μL of MTT (5 mg/mL) were added to the culture for 4 h. Then, after addition of 110 μL of solubilization solution (10% SDS in HCl 0.01 M), cells were incubated at 37 °C overnight. Specific optical density was measured at 540 nm, using 690 nm as reference wavelength in the microreader SLT-Lab (Salzburg, Austria). For the proliferation studies, each experiment was repeated three times. Each treatment was performed with eight replicates. Student’s test and multi factorial ANOVA were used to evaluate statistical differences among treatments.

### 3.6. Transient Transfection Assay

Experiments were done as described in [14,18,21], with the appropriate change. Transfections were performed in 6-well plates, when cells were 80% confluent (approximately 3 × 10^5^ cells) on the day of transfection. In addition, 5 μg of pEGFP-C1 DNA and the right amount of 2 mM stock solution of GP16_6, FGP6 and FGP8 were added to 2 mL of serum-free DMEM medium (Dulbecco’s modified Eagle medium, Gibco), supplemented with 50 µg/mL of gentamycin (Gibco) at final concentrations of 20, 10, 5, and 2.5 μM, mixed rapidly and incubated at room temperature for 20 min. Furthermore, 500 μL of each mixture were carefully added to the cells following the aspiration of the culture medium. The same experiments were repeated by adding DOPE to the plasmid-surfactant mixture at a surfactant: DOPE molar ratio of 1:2. Lipofectamine LTX (Invitrogen, Walthman, MA, USA) transfection reagent was used as a positive transfection control, and the transfection was prepared as suggested by the manufacturers [14,18,21]. 

Mixture and cells were incubated at 37 °C in a humidified atmosphere of 95% air/5% CO_2_ for 6 h. After that, 1 mL of cEMEM (Gibco), supplemented with 10% FBS, was added to each transfected well and left to incubate for 48 h. Transfected cells were observed under a fluorescence microscope for EGFP expression. Five random fields were examined from each well, and each experiment was repeated three times. Student’s *t*-test and multi factorial ANOVA were used to evaluate statistical differences among treatments.

### 3.7. Sample Preparation and Atomic Force Microscopy (AFM) Imaging 

DNA samples were prepared as described in [18,21]. The plasmid DNA was diluted to a final concentration of 0.1 mM in deposition buffer (4 mM Hepes, 10 mM NaCl, 2 mM MgCl_2_, pH = 7.4) either in the presence or in the absence of GP16_6, FGP6 and FGP8 at the same *N*/*P* ratio (ratio between negative and positive charges) as transfection. The mixture was incubated for 5 min at room temperature; then, a 20 μL droplet was deposited onto freshly cleaved ruby mica (Ted Pella, Redding, CA, USA) for one minute. The mica disk, rinsed with milliQ water, was dried with a weak stream of nitrogen. AFM imaging was performed on the dried sample with a Park XE-100, Suwon, Korea, operating in tapping mode. Commercial diving board silicon cantilevers (NSC-15 Micromash Corp., Sofia, Bulgaria) were used. The images were optimized by the software XEI (Park Systems, Suwon, Korea). 

### 3.8. Dynamic Light Scattering (DLS)

We performed DLS size distribution analysis (Zetaplus, Brookhaven Instrument Corporation, Holtsville, NY, USA) operating at the wavelength λ = 658 nm at the fixed temperature T = 25 °C. Samples were prepared at 200 μM and 20 μM concentrations of *gemini* fluorinated surfactants with plasmid DNA at *N*/*P* ratio = 1. Samples (2 mL) of each solution were placed in disposable polystyrene cuvettes for DLS measurements. Data were analyzed by BIC Dynamic light Scattering Software (Brookhaven Instruments: software version 3.34) and by our own software. 

## 4. Discussion

As we stated in the introductive section, *gemini* surfactants could be a very good candidate as non-viral vectors in gene delivery, due to their multiple cationic charge, necessary for binding and compacting DNA, and to their superior surface activity. 

We have obtained interesting results both by using very simple bisquaternary ammonium *gemini* surfactants, derivatives of *N*,*N*-bisdimethyl-1,2-ethanediamine (bis-CnBEC) [14] and dipyridinium *gemini* surfactants [14,21], when formulated with DOPE [L-phosphatidylethanolamine dioleoyl (C18:1,[cis]-9)]. We were amazed by the tight relationship we have found between solution thermodynamics of dipyridinium gemini surfactants and their DNA compacting ability, making “old” macroscopic thermodynamic methods essential to rationalize their biological behavior at a molecular level [14,16,17,18,21]. 

More recently, fluorinated and partially fluorinated *gemini* surfactants (otherwise called “hybrid surfactants”) have also been proposed for transfection purposes, due to the chemical and biological inertness, their lower acute toxicity and lower hemolytic activity compared to their hydrogenated counterparts [46,47]. Fluorine atoms have lower polarizability and larger van der Waals radii compared to hydrogen atoms. In addition, the fluorocarbon chains have a rigid rod-like shape with a period of twist of 13 carbon atoms, while the hydrocarbon chains show the well-known zig-zag structure [48,49]. These structural aspects and the strength of the C-F bond explain their chemical and biological inertness. Partially fluorinated *gemini* surfactants confirm the tight relationship between their solution thermodynamics and biological performance [17,21].

Moreover, if the polar head is constituted by a heterocyclic cationic moiety, the possibility of intercalation with the DNA bases could enhance their compacting ability [3,31,39]. Some terms of the homologous series of *gemini* bispyridinium surfactants—both hydrogenated and highly fluorinated—have shown a very interesting gene delivery ability, superior to that of the commercial reagent generally used for transfection experiments, when formulated with a helper lipid such as DOPE. We have found that this ability is strictly related to the spacer length, and we were able to show for the first time that their transfection activity is related to their unexpected thermodynamic properties in solution [16,17,18,21]. In fact, in the attempt to determine a group contribution for the apparent and partial molar enthalpies in solution of the methylene group when this group is added to the spacer, we noted that, at a given length of the spacer, the behavior in solution dramatically changed, not allowing for the evaluation of the group contribution. The interest in determining a group contribution is related to the possibility to evaluate thermodynamic properties in solution of new designed surfactants by calculation, without experimental measurements. On the contrary, the group contribution additivity for the counterion is respected, independently on the spacer length, as in the case of monomeric surfactants [50,51], showing that this peculiarity is due to the structure of the molecule itself. The deviation of the properties from those theoretically predicted could suggest a phase transition or a change in conformation of the molecule in solution. We explained the unexpected behavior of *gemini* bispyridinium surfactants, by a conformation change of the molecule determined by stacking interactions between the two pyridinium rings, appearing at an optimum length of the spacer. These interactions give rise to a sort of molecular tongs able to grip basic DNA groups near each other. The compounds studied till now deviating from the predicted trend of the apparent and molar enthalpies vs. concentration show the greatest gene delivery ability. This happens for the compound with spacer four carbon atoms long for protiated compounds and eight atoms long for highly fluorinated compounds. With the aim of optimizing the gene delivery ability of the compounds under study, we decided to synthesize the compounds both hydrogenated and highly fluorinated having a spacer length of six carbon atoms, to fill the gap between the four and eight methylene spacers previously studied and showing the best performances. The sixteen carbon atoms hydrophobic chains length of the hydrogenated compound was chosen because this length is generally the most biologically active for the hydrogenated compounds [52]. On the other hand, the hydrophobic chains of the partially fluorinated *gemini* surfactant are shorter because the hydrophobicity of a -CF_2_- group is about 1.5 times that of a -CH_2_- group, as shown by the values of the respective critical micelle concentration (cmc) [50,51].

Having in mind a possible application of the compounds as non-viral vectors in gene-based medicine particularly for treating cystic fibrosis associated pulmonary disorders, we have tested the new synthesized compounds and FGP8 not only on the RD4 cell line but also on murine pulmonary adenoma (LA-4) and human pulmonary adenocarcinoma A549 cell lines. A549 cells are adenocarcinomic human alveolar basal epithelial cells, used as models for the study of lung cancer and for the development of drug therapies against it. We are used to testing transfection ability on RD4 cells [14,18,21] because they are easy to handle, fast growing and a good compromise between very difficult to transfect cells and very easy to transfect cells with traditional methods (electroporation, lypofection and calcium phosphate precipitation). Moreover, they are derived from a nasty human cancer, so that the data obtained from the molecules employed in the present paper could be relevant for oncologic gene transfer studies. When genes are delivered to the respiratory or to the biliary epithelium, there is another barrier to overcome, due to the presence in their biological fluids of endogenous hydrogenated surfactants, as pulmonary surfactants, or bile salts, able to destroy the lipoplexes. The use of vectors having highly fluorinated alkyl chains, hydrophobic and lipophobic at the same time, may protect the lipoplexes from unwanted interactions with the biological medium [40,41,42]. 

The interaction with DNA of FGP6 and FGP8 seems to not be hindered by the presence of hydrophobic and lipophobic fluorinated moiety. In fact, the positive charges are delocalized on the two pyridinium rings in a hydrogenated environment, constituted by the hydrogenated spacer and by the two -CH_2_-CH_2_- groups (see Figure 1), connecting pyridinium rings with the fluorinated alkyl chains.

The formation of lipoplexes is a necessary but not sufficient condition for obtaining gene delivery inside the cells. The results of the gene delivery experiments on RD-4 cells (Figure 5, left) showed that hydrogenated GP16_6 is not able to give rise to transfection, in agreement with the AFM images indicating its inability to compact DNA. 

On the other hand, nanoparticles obtained by partially fluorinated compounds alone (Figure 5c,d) are also unable to deliver genes inside the cells, whereas, if transfection experiments are carried out in the presence of the helper lipid dioleoylphosphatidylethanolamine (DOPE), the percentage of transfected cells (the green cells expressing EGFP in the fluorescence microscope images in Figure 5, left) becomes surprisingly high, superior to those of the commercial transfection agent, LTX, used as positive control. It is known that DOPE enhances the transfection activity of cationic formulations through the stabilization of DNA/lipid complex and facilitates the transfer of DNA in the context of endosomal escape [53,54].

The presence of DOPE is necessary to obtain transfection with partially fluorinated surfactants, the fluorinated moieties being both hydrophobic and lipophobic. This fact implies the interaction of the fluorinated chains in a hydrogenated environment, i.e., the fluorinated moieties tend to stick together to avoid aqueous and hydrogenated environments. In this way, the molecule adopts the shape most idoneous for compacting DNA, giving rise to smaller and more compact DNA nanoparticles. At the same time, it is necessary for the fusogenic activity of DOPE to overcome the repulsion between the fluorinated DNA nanoparticles and the hydrogenated phospholipids constituting the cell membrane. The obtained results confirm what we have previously observed [21]: in the presence of DOPE, all the partially fluorinated compounds, except for FGP12, are able to deliver genes inside RD4 cells, with an efficiency increasing in the order FGP4 < FGP3 << FGP6 ≅ FGP8 (see Figure 7). It must be outlined that FGP6 and FGP8 both give rise to a remarkable transfection efficiency with a difference that is not statistically relevant.

In Figure 7a, the gene delivery ability of GP16_*n* hydrogenated *gemini* surfactants and of partially fluorinated *gemini* surfactants FGP*n* (Figure 7b) are shown as a function of the spacer length, *n*, for the experiments done only with the surfactants and with surfactant: DOPE ratio = 1:2.

When hydrogenated bis-pyridinium *gemini* surfactants are considered, only the compound with spacer constituted by four methylene groups can give rise to a transient transfection, interestingly both with and without DOPE. We have suggested, considering the information from thermodynamic data, that the compound with the spacer formed by four carbon atoms is able to form a kind of molecular tongs able to grip basic DNA groups near each other. The spacer is long enough to allow for the formation of the tongs but short enough to maintain the hydrophobic tails that are quite close to interact each other, thus allowing the formation of compact and near spherical DNA particles [18]. 

For the fluorinated compounds (Figure 7b), a greater length of the spacer is needed for molecular tongs formation because of the steric modification induced by fluorination. In fact, fluorine atoms have a larger van der Waals radius and a lower polarizability than the hydrogen atoms [42,43]. It is not surprising that the best transfection ability is shown by the compounds that are 6–8 carbon atoms long.

When the length of the spacer increases, this arrangement becomes more difficult and the hydrophilic pyridinium rings tend to stay quite far apart and also interact with the DNA bases far from each other, forming structures not suited for gene delivery. This is the case of the spacer with 12 carbon atoms. Moreover, the toxicity of the fluorinated compounds seems to be spacer length dependent, increasing with the length of the spacer (see also [21]).

When pulmonary cells are tested, a Pei transfection reagent was used as a positive control, as shown in Figure 5, because the LTX standard reagent was too toxic. The compounds tested, neither hydrogenated nor partially fluorinated, were able to transfect human A459 (see Figure 5, right) or murine LA4 (not shown), notwithstanding the great performance of FGP6 and FGP8 on RD4 cells. It seems that pulmonary cells are harder to transfect and more sensitive to the vector used, as their behavior towards LTX standard reagent outlines. Therefore, at first sight, they do not seem suitable for gene therapy treatment of cystic fibrosis, but they can find applications in other fields, due to their high gene delivery ability when formulated with DOPE.

## 5. Conclusions

Gene-based medicine has become a clinical reality. The possibility to prevent infectious diseases by genetic immunization has focused the scientific and economic efforts of the pharmaceutical industry and of the governments to fight the pandemic emergency determined by the spreading worldwide of the SARS-CoV-2 virus with the use of vaccines based on DNA or mRNA. Regardless of the type, the genetic material must be delivered inside the cells by means of vectors. Due to the risk of adverse or immunogenic reactions, or replications connected with the more efficient viral vectors, the development of very efficient and safe vectors is a landmark for gene-based medicine. In particular, DNA vaccines, till now formulated using viral vectors, could benefit from very efficient non-viral vectors, able to protect the genetic material in its way toward the nucleus. *Gemini* cationic surfactants could be good candidates as non-viral vectors, taking advantage from their multiple charges. The study of *gemini* bis-pyridinium surfactants, both hydrogenated and fluorinated, confirms that pyridinium *gemini* surfactants could be a valuable tool for gene delivery purposes. Because their performance highly depends on the spacer length and is strictly related to their structure in solution, here we have synthesized and studied the new terms with a spacer that is six methylene long, both hydrogenated and fluorinated. All the fluorinated compounds can compact DNA and deliver a plasmid carrying an EGFP expression cassette on RD-4 cells, when co-formulated with DOPE in a 1:2 ratio. The fluorinated compounds with a spacer formed by six (FGP6) and eight carbon atoms (FGP8) give rise to a very interesting gene delivery activity, superior to that of the commercial reagent This means that the length of the spacer appropriate for activity fall in a broader, but always limited, range of values. In fact, the spacer that is 12 carbon atoms long is inactive for both classes of compounds. On the other hand, the hydrogenated compound GP16_6 is unable to sufficiently compact the DNA, as shown by AFM images. In the hydrogenated series, only the compound with a spacer that is 4 carbon atoms long delivers the genetic material with significant efficiency, both with and without DOPE, probably because this length of the spacer allows the molecule to assume the right configuration for intercalating with DNA, forming a sort of molecular tong. If the hydrophobic chain is modified by partial fluorination, a greater length of the spacer is needed for the folding of the molecule because of the greater volume and rigidity of the fluorinated moiety, but the presence of DOPE is essential to overcome the repulsion between the fluorinated chains and the phospholipids of the cell membrane. Verifying the effect of the hydrophobic chain length, particularly on fluorinated compounds, to optimize their efficiency still needs to be completed.

## Figures and Tables

**Figure 1 ijms-23-03062-f001:**
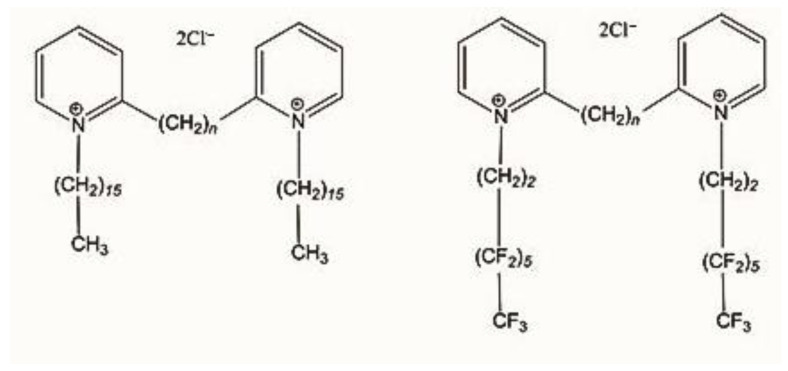
Compounds under study, *n* is the number of carbon atoms in the spacer.

**Figure 2 ijms-23-03062-f002:**
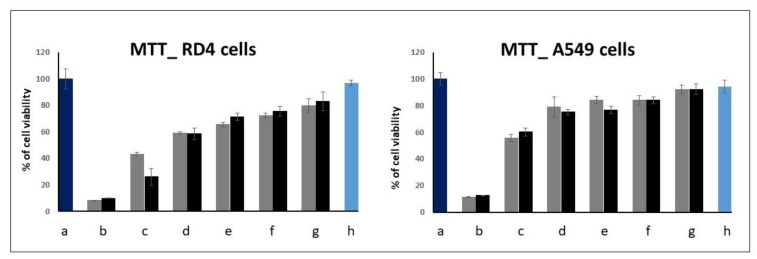
MTT test on RD4 (left) and A549 cells (right) for GP16_6: (**a**) untreated cells; (**b**) 40 µM; (**c**) 20 µM; (**d**) 10 µM; (**e**) 5 µM; (**f**) 2.5 µM; (**g**) 1.25 µM; (**h**) only DOPE. In gray, GP16_6 alone; in black, GP16_6: DOPE = 1:2. Values are the mean ± S.D. of three independent experiments (*n* = 8 per treatment, *p* < 0.05).

**Figure 3 ijms-23-03062-f003:**
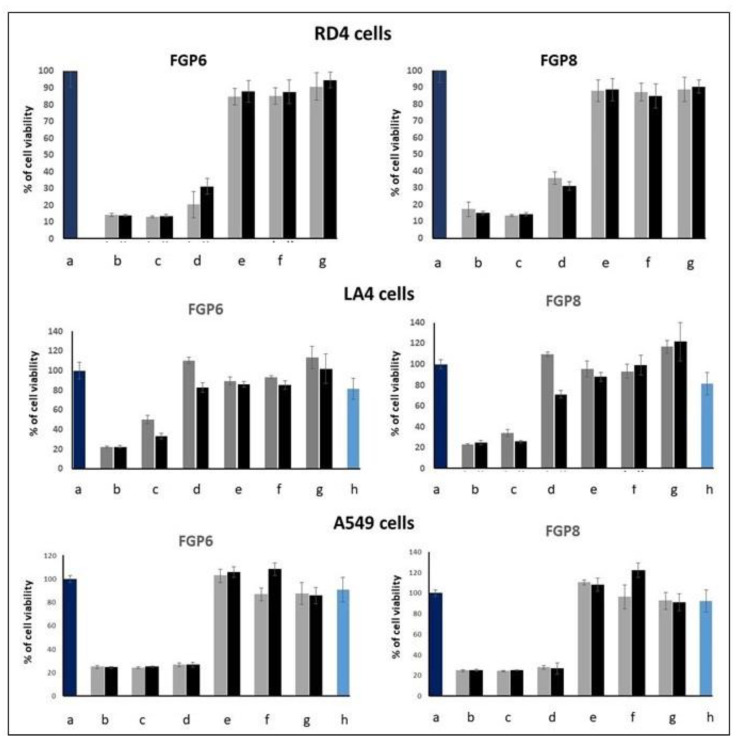
Effects of FGP6 (**left**) and FGP8 (**right**) on RD4 (**top**), LA4 (**middle**) and A549 (**bottom**) cell lines proliferation measured by MTT test after 48 h of incubation: (**a**) untreated cells; (**b**) 40 µM; (**c**) 20 µM; (**d**) 10 µM; (**e**) 5 µM; (**f**) 2.5 µM; (**g**) 1.25 µM; (**h**) only DOPE. In gray, FGP*n* alone; in black, FGP*n*: DOPE = 1:2. Values are the mean ± S.D. of three independent experiments (*n* = 8 per treatment, *p* < 0.05).

**Figure 4 ijms-23-03062-f004:**
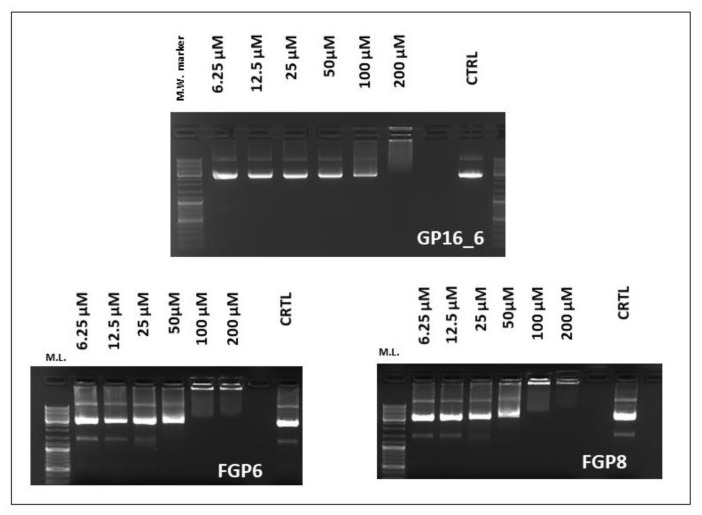
EMSA experiments showing complexation of GP16_6, FGP6 and FGP8 with circular plasmid pEGFP-C1. Shifting is observable as a function of concentration (μM). As a negative control, only the plasmid was used, which is completely unshifted. 100 μM cationic lipid concentration corresponds to *N*/*P* ≈ 1.

**Figure 5 ijms-23-03062-f005:**
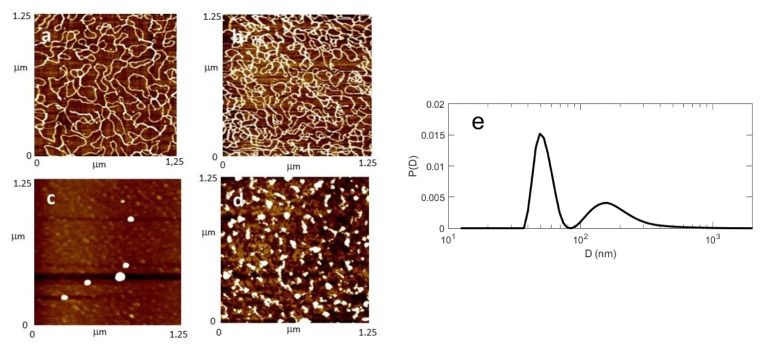
(**a**–**d**) are AFM images showing the effect induced on DNA plasmid by incubation with the compounds under study: (**a**) plasmid alone; (**b**) plasmid + GP16_6; (**c**) plasmid + FGP6; (**d**) plasmid + FGP8. All images were obtained with supercoiled 0.5 nM pEGFP-C1 plasmid deposited onto mica and with the microscope operating in tapping mode in air at the same *N*/*P* ratio as transfection. For image d, see Ref. [21]. (**e**) DLS volume distribution of plamid + FPG6, the same sample as (**c**).

**Figure 6 ijms-23-03062-f006:**
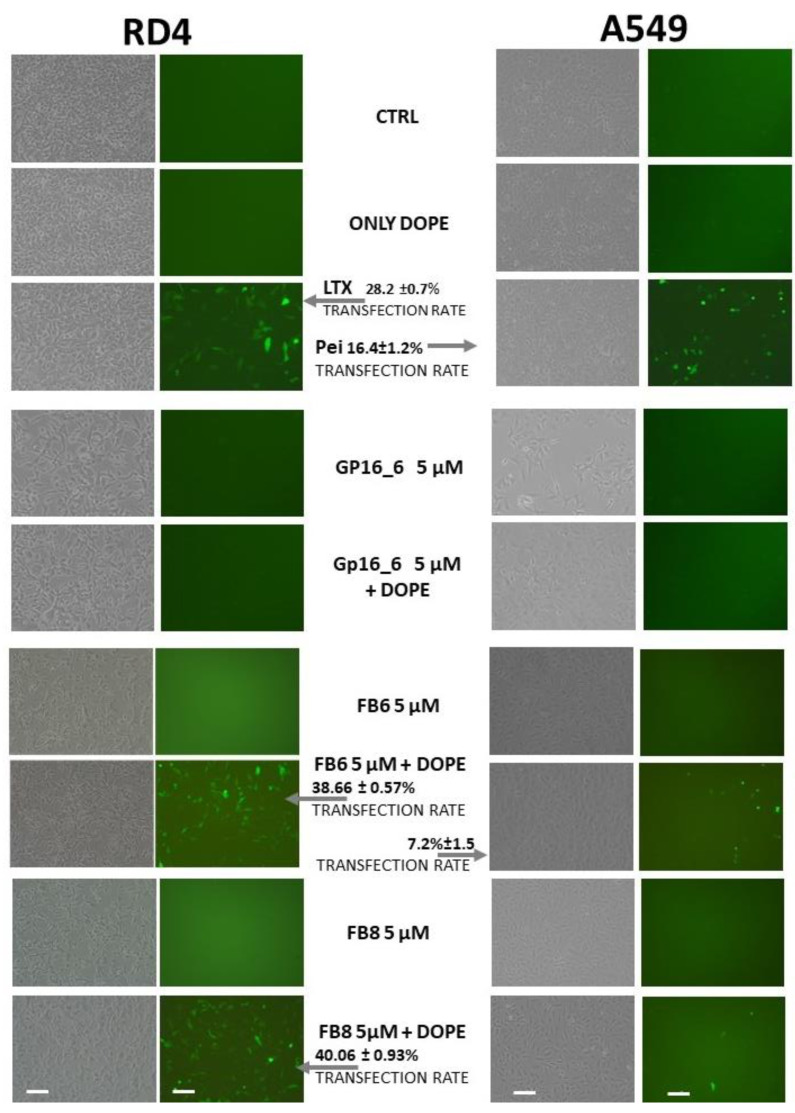
Transfection of RD4 (**left**) and A549 cells (**right**) by GP16_6, FB6 and FB8 5 μM. For each kind of cells, on the left, phase contrast and, on the right, fluorescence microscope observation of the transfected cells (as shown by green cells expressing EGFP) is shown. The experiments were done with the surfactant alone and with surfactant:DOPE = 1:2. Cells are not transfected with DOPE alone (see [14,18,21]). Transfection rate of FB8 5 μM + DOPE is not indicated because it is <1%. The scale bars = 20 μm are common for all of the micrographs.

**Figure 7 ijms-23-03062-f007:**
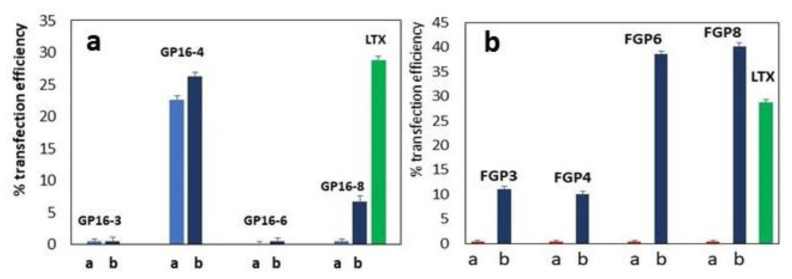
Gene delivery ability on RD4 cells of GP16_*n* hydrogenated *gemini* surfactants and partially fluorinated *gemini* surfactants FGPn as a function of the spacer length, *n*: (**a**) experiments done only with the surfactants and (**b**) with surfactant:DOPE ratio = 1:2. Positive control by a commercial reagent is in green. Spacer 12 (not shown) is unable to deliver DNA. See [18,21].

## Data Availability

Not applicable.

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
