# Peer review of "Gene-Delivery Ability of New Hydrogenated and Partially Fluorinated Gemini bispyridinium Surfactants with Six Methylene Spacers"

_ijms, 2022, doi:10.3390/ijms23063062_

Round 1

Reviewer 1 Report

The paper describes the synthesis of hydrogenated and partially fluorinated gemini surfactants having six methylene spacers. This is a merely extension of the sudies already performed on very similar surfactants (see references). I do not see enough novelty which deserves publication in this journal. Moreover, characterization of the novel compounds lacks for the 13C NMR and, possibly, 19F NMR.

Reviewer 2 Report

The manuscript ijms-1602242 "Gene-delivery ability of new hydrogenated and partially fluorinated Gemini Bispyridinium surfactants with six methylene spacers" by Fisicaro and co-workers describes the synthesis hydrogenated and partially fluorinated dipyridinium gemini surfactants with chlorides as counterions and the study of their transfection ability in RD-4, A549, LA-4 cells. The authors have interesting experimental results, so I believe that this paper will be of interest to the readers of IJMS.

Questions and comments:

1) What is the polydispersity index (PDI) of the supramolecular system in Figure 5e? Why didn't the authors measure the zeta potential?

2) Why do the authors not study the morphology and size of lipoplexes based on a mixture of gemini surfactants and DOPE? This system seems to be the most promising for transfection.

3) All obtained new compounds should be characterized by 1H, 13C, 19F NMR, IR spectroscopy and mass spectrometry. Images of all spectra should be in supplementary materials.

4) The authors do not indicate the instrument and experimental conditions for NMR spectroscopy and elemental analysis.

5) Since such studies have been carried out for a long time, including by a team of authors, I recommend comparing the results obtained in this work with the previous ones.

6) The manuscript contains many errors (see some below). Please double-check it.

Line 134 – CaCl2.

Lines 145, 158, 168 – 1H-NMR.

Line 170 – C48H86Cl2N2.

Lines 275, 294 – 2,5 μM

Figure 6 should be after the text in section 3.4, not before it.

Reviewer 3 Report

The manuscript is well written and all the claims are supported by sufficient experimental data. The manuscript can be published after minor revision:

-The characterization of the DDS is poor. Can we add FTIR or Raman spectra to further prove the loading RNA into the surfactant? What about mass spectrophotometer?

-Figure 2 and Figure 3, the x-axis should be the concentrations. 

Round 2

Reviewer 2 Report

I thank the authors for answering my questions and improving the manuscript.

Please add the instrument and experimental conditions for IR spectroscopy, and a description of the IR spectra of each synthesized compounds (Figures S4, S5) in part 2.1.1.